# Transcriptome Analysis of Early Senescence in the Post-Anthesis Flag Leaf of Wheat (*Triticum aestivum* L.)

**DOI:** 10.3390/plants11192593

**Published:** 2022-10-01

**Authors:** Ling Lei, Dan Wu, Chao Cui, Xiang Gao, Yanjie Yao, Jian Dong, Liangsheng Xu, Mingming Yang

**Affiliations:** 1College of Agronomy, Northwest A&F University, Xianyang 712000, China; 2Xinyang Normal University, Xinyang 464000, China; 3Chongqing Academy of Chinese Meteria Medica, Chongqing 400000, China; 4Wheat Engineering Research Center of Shaanxi Province, Xianyang 712000, China; 5College of Plant Protection, Northwest A&F University, Xianyang 712000, China

**Keywords:** wheat, transcriptome, early senescence, flag leaf, post-anthesis

## Abstract

Flag leaf senescence is an important determinant of wheat yield, as leaf senescence occurs in a coordinated manner during grain filling. However, the biological process of early senescence of flag leaves post-anthesis is not clear. In this study, early senescence in wheat was investigated using a high-throughput RNA sequencing technique. A total of 4887 differentially expressed genes (DEGs) were identified, and any showing drastic expression changes were then linked to particular biological processes. A hierarchical cluster analysis implied potential relationships between *NAC* genes and post-anthesis senescence in the flag leaf. In addition, a large set of genes associated with the synthesis; transport; and signaling of multiple phytohormones (JA, ABA, IAA, ET, SA, BR, and CTK) were expressed differentially, and many DEGs related to ABA and IAA were identified. Our results provide insight into the molecular processes taking place during the early senescence of flag leaves, which may provide useful information in improving wheat yield in the future.

## 1. Introduction

Wheat (*T**riticum*
*aestivum* L.) is a worldwide staple food crop. It is grown on about 200 million hectares across diverse global environments [1], and it is consumed by 2.5 billion people in 89 countries, accounting for 20% of the global caloric intake (Centro Internacional de Mejoramientode Maizy Trigo (CIMMYT)). CIMMYT has predicted that 70% more wheat will be required by 2050 to continue supplying a similar proportion of the global population. To achieve this goal, many challenges from physiological, agronomic, socioeconomic, environmental, and managerial aspects need to be overcome.

The grain yield of wheat is determined by the successful formation, partitioning, translocation, and accumulation of assimilates during grain filling. Assimilates mainly come from two sources [2]. First is the accumulation of assimilates formed by photosynthesis; approximately 70–90% of the overall grain yield is supplied by photoassimilates [3]. Leaves, being the major site of photosynthetic activity, play an important role in determining the grain yield of wheat. Flag leaves, compared with other leaves, contribute the most photosynthates (41–43%) [4]. It has been reported that about 60% of grain saccharides are derived from the photosynthates of the flag leaf [5]. Second, the remobilization of pre-anthesis assimilates stored in vegetative organs. With the senescence of vegetative organs, the structure, metabolism, and gene expression of cells have undergone a large number of orderly changes, accompanied by the transport of the degradation products of proteins, lipids, and nucleic acids to the reservoir tissue. The disintegration of the photosynthetic apparatus is a major event during senescence, mainly as ribulose-1,5-bisphosphate carboxylase/oxygenase (Rubisco) [6]. With the hydrolysis of Rubisco, a large amount of free amino acids will be produced and transported to the grain [7]. Photosynthetic organelle chloroplasts are the main organelles for iron storage (80%), and a large number of Fe released after the disintegration of the organelle will be reused by the grains, so that the Fe content in the leaves decreases by nearly 10-fold when the grains mature [8]. In addition, up to 75% of the reduction in nitrogen in leaves is from the chloroplasts [9], which provide most of the nitrogen that is redistributed to wheat grain [10]. The redistribution of N is also found in other crops, such as 50–90% of the N in rice ears [11] and 60–80% of the N in corn kernels [12] from N transport in leaves. Therefore, leaf senescence occurs in a coordinated manner during grain filling. Premature aging will affect the formation of assimilates, and delayed aging will affect the optimal time for nutrients to be transferred to reproductive organs. The senescence time of a flag leaf is one of the important factors affecting the crop yield and grain mineral element content.

Aging is affected by many internal and external factors. The external factors include cultivation conditions, such as temperature, sunshine, water and nutrition supply, diseases and pests, etc. The internal factors include the developmental stage and hormone level. Although various factors are interrelated and influence each other, the internal factors play a leading role. Different techniques have been used to identify many senescence-associated genes (SAGs), which are mainly involved in degradation, biosynthetic and regulatory processes, stress responses, and transport [13,14,15,16]. In SAGs, NAC (no apical meristem (NAM), ATAF1/2, and cup-shaped cotyledon (CUC)) and WRKY transcription factors (TFs) play important roles in regulating gene expression changes during leaf senescence. For example, delaying senescence by declining the expression of *NAM-B1* reduced the contents of protein, zinc, and iron in wheat grain by more than 30% [13,17]. The *NAM*-1 allele was also related to delaying senescence and prolonging the grain-filling period of wheat [18]. *ZmNAC126* could be induced by ethylene, the ectopic overexpression of *ZmNAC126* in Arabidopsis, and maize enhanced chlorophyll degradation and promoted leaf senescence [19]. The ABA signaling pathway and antioxidant enzyme systems are involved in *TaNAC29*-mediated stress tolerance mechanisms and plays an important role in the senescence process [20]. Expression of the *TaSNAC11-4B* gene was induced by all the senescence-promoting phytohormones, namely abscisic acid (ABA), jasmonic acid (JA), ethylene, and salicylic acid (SA) [21]. EIN3 (a key transcription factor in the ethylene signaling pathway) and ORE1/NAC2 accelerate chlorosis during ethylene-mediated leaf senescence by directly activating chlorophyll catabolic genes in Arabidopsis [22]. *TaWRKY13-A*, *TaWRKY40-D,* and *TaWRKY42-B* have also been reported to be involved in the regulation of leaf senescence and related to the JA or ABA pathway [23,24,25]. In conclusion, NAC and WRKY transcription factors seem to be inextricably linked with plant hormones, such as ABA, JA, and ethylene, and play crucial roles in senescence. In addition, it has been reported that the interaction between different hormones is also related to leaf senescence [26].

Although there are many studies on senescence in wheat, Arabidopsis, rice, maize, etc., most of them are based on the regulation of a single gene. Using 9K probe sets that were designed against unigenes from 35 wheat tissue cDNA libraries, Gregersen et al. (2006) [27] conducted a microarray experiment on the transcriptome of a wheat flag leaf during its senescence to obtain information about the senescence process. However, Poole et al. (2007) [28] pointed out the insufficiency of commercial and homemade microarrays used in wheat studies and found no comparability existed among these results. Moreover, the huge size, polyploid complexity, and estimated 75% repetitive DNA sequence content of the wheat genome [29] also largely limited their use. The relationship between flag leaf photosynthesis, the timing of senescence, and grain filling are not clear [30]. In recent years, with the development of high-throughput sequencing technology, transcriptome sequencing (RNA-seq) has become a powerful tool for studying complex biological processes and identifying candidate genes related to specific biological functions at the molecular level. It not only contains genomic information but also reveals the flow of biological information to cells in the genome and can find unknown transcripts, fusion genes, and genetic polymorphisms, which is impossible for chips. Its high sensitivity will also have a better detection of low abundance transcripts. The availability of a large number of transcriptome datasets provides an opportunity to comprehensively analyze the senescence process of wheat. Therefore, in this study, we used RNA-seq technology and the model wheat variety “Chinese Spring” that was sequenced by the whole genome to characterize possible senescence-related genes at different development stages of flag leaves of wheat, an important food crop. The present research extends the wheat cDNA array analysis of Gregersen and Holm (2006) [27] by examining the expression of extensively annotated wheat genes and further separately analyzing the substance translocation, transcriptional regulation, and phytohormone signaling during leaf senescence. It contributes to a better regulation understanding of light assimilate formation and the redistribution of senescence materials and identifies the genes involved in early leaf senescence, further providing the possibility for the simultaneous improvement of wheat yield and quality.

## 2. Materials and Methods

### 2.1. Plant Material

The bread wheat cultivar Chinese Spring was field grown at the Wheat Breeding Center of Northwest A&F University in Yangling, Shaanxi Province (34°26′ N, 108°14′ E) at an altitude of 527 m. It belongs to a warm temperate monsoon climate, with an annual average temperature of 12.9 °C, average annual precipitation of 500–700 mm, and a frost-free period of 211 d. During the wheat flowering until harvest (15 April–2 June), the daily maximum temperature is 33 °C, the daily average temperature is 10–26 °C, the average relative humidity is 66.87%, and the daily average relative humidity is 40.67–97.31% (Appendix A). Meteorological data were from https://rp5.ru/ (accessed on 15 September 2022). The amount of fertilizer applied before sowing was 120 kg/ha urea and 300 kg/ha diammonium phosphate. The sowing amount of wheat seeds was 150 kg/ha, and row spacing was 0.20 m, the plot area 35 m^2^ (5 m × 7 m). Irrigation was carried out at the jointing stage of wheat, and 225 kg/ha of urea was applied. The main culm spikes were tagged upon anthesis, and the flag leaves were sampled at 0 (20 April), 15 (5 May), 25 (15 May), and 30 (20 May) days after anthesis (DAA). The weather conditions on the day of sampling are shown in Appendix A. At 6 p.m. on each date, using the five-point sampling method, 10 flag leaves were collected from different plants. Samples were snap frozen in liquid nitrogen and stored at −80 °C for later use.

### 2.2. RNA Extraction and cDNA Library Preparation

Total RNA was extracted using an RNA extraction kit, following the manufacturer’s protocol (Biotech, Beijing, China). The integrity, quantity, and purity of each RNA sample were examined by 1% agarose gel electrophoresis using an Agilent 2100 Bioanalyzer (Agilent Technologies, Santa Clara, CA, USA) and a NanoDrop^TM^ 1000 (Thermo Fisher Scientific, Waltham, MA, USA). Ten equimolar high-quality RNA samples for each time point were mixed and sent to Novogene Co., Ltd. (Beijing, China) for cDNA library preparation. Finally, four libraries were sequenced on the Illumina HiSeq 2500 platform for paired-end sequencing with a read length of 125 bp (Novogene, Beijing, China).

### 2.3. Data Preprocessing and Read Mapping

Raw data were first preprocessed by filtering adaptor contaminants and reads with ≥10% poly (N), as well as ≥50% low-quality bases (SQ ≤ 5). The remaining data were utilized for subsequent analyses. The reference genome of the bread wheat cultivar Chinese Spring used for mapping was produced by the International Wheat Genome Sequencing Consortium (ftp://ftp.ensemblgenomes.org/pub/current/plants/fasta/triticum_aestivum/dna/Triticum_aestvum, accessed on 9 June 2015; IWGSP2.25.dna.toplevel.fa.gz/, accessed on 9 June 2015). TopHat v2.0.12 [31] was applied, where all clean reads were aligned against the reference genome, and no more than two nucleotide mismatches were allowed. Subsequently, the reads that simultaneously mapped to multiple transcripts were eliminated, and only the unambiguously mapping reads were retained. Meanwhile, novel transcripts were constructed and identified by the reference annotation-based transcript (RABT) assembly method that was built on Cufflinks v2.0.2 with default parameters [32].

### 2.4. Identification of DEGs

Uniquely mapped reads were counted in union mode with HTSeq v0.6.1 [33]. The RPKM (reads per kilobase of exon model per million mapped reads) value was calculated as the expression amount of the transcript. Prior to normalization, only the transcripts that were found in at least two libraries and more than three mapped reads were kept. The differential expression analysis was conducted in a timepoint-wise manner by employing DEGSeq R package v1.12.0 [34] after all read counts were normalized by the TMM method [35]. Whether the gene has significant differential expression was evaluated by the difference multiple of expression and the *p*-value corrected by the Benjamini and Hochberg method [36]. When the expression difference of a gene in the two libraries was more than 2-fold (|log2[foldchange]| > 1) and the corrected *p*-value was less than 0.005, the gene was considered a significantly differentially expressed gene (DEGs).

### 2.5. Cluster Analysis of DEGs

Heat map clustering of DEGs was performed using the heatmap.2 function of Bioconductor package v 1.0.1 ggplots (Clustering mode: hclust; Distance metric: Pearson correlation; linkage method: average). At the same time, the relative expression levels of the DEGs were calculated using Multi Experiment Viewer v4.7.4, DBF_L1 samples as controls, and the expression trend clustering model of these genes was established using the same hierarchical clustering parameters.

### 2.6. GO and KEGG Enrichment Analysis

Using a Hidden Markov model (HMM) of protein families from the TIGR (http://blast.jcvi.org/web-hmm, accessed on 15 June 2015) and PFAM databases (http://pfam.xfam.org, accessed on 15 June 2015), differentially expressed novel transcripts were annotated by HMMscan. The parameters that controlled the results were based on a previous study [37]. The annotations for other DEGs and the enrichment analysis were subsequently conducted with GOseq R package v1.20.0 based on Wallenius’ noncentral hypergeometric distribution. KOBAS v2.0 was utilized to identify biological pathways in which DEGs were involved and calculate the statistical significance of each pathway using the default criteria [38]. The GO terms and KEGG pathways with corrected *p*-values of less than 0.05 were considered significantly enriched.

### 2.7. Identification of Senescence-Associated Genes

All DEGs were blasted against protein sequences from the Leaf Senescence Database (http://www.eplantsenescence.org, accessed on 16 June 2015) and the Arabidopsis Phytohormone Database (http://ahd.cbi.pku.edu.cn, accessed on 16 June 2015) by employing BLASTx (E value ≤ 1 × 10^−10^). In addition, the differentially expressed new genes were compared with the NCBI nonredundant protein database (NR) (BLASTx, E value ≤ 1 × 10^−10^), and the protein sequences with the highest degree of similarity obtained and the protein sequences of other DEGs extracted from Enesembl Plant were analyzed to identify the transcription factors in them; at the same time, the hidden Markov model of all transcription factors in Pln TFDB was used to analyze the possible transcription factors and their types in all DEGs with HMMER3.0 (http://hmmer.janelia.org/, accessed on 16 June 2015). After that, all the above identification results were confirmed with the NCBI tool BLASTn. Finally, expression clustering was performed according to the clustering analysis of DEGs.

### 2.8. Quantitative Real-Time PCR

Total RNA extraction and quality detection were performed as described in Section 2.2, and EasyScript^®^ First-strand cDNA synthesis Supermix (TRAN, Beijing, China) was used for cDNA synthesis. Using a QuantStudio^TM^ 7 FleX Real-Time PCR Detection System (Thermo Fisher Scientific, Waltham, MA, USA), qRT-PCR was done using the PerfectStart^®^ Green qPCR Supermix (TRAN, Beijing, China). The qRT-PCR was completed in the three biological replications where the same samples were used for RNA-seq analysis. To standardize the relative expression, the actin gene was used as a housekeeping gene. The relative expression values for each of the 12 selected genes were obtained by the delta–delta threshold cycle (2^−ΔΔCt^) method. The primers used in the qRT-PCR analysis are listed in Appendix A. The relative expression data were subjected to ANOVA in SPSS (version 22, SPSS Inc., Chicago, IL, USA). The correlation between RNA-seq and qRT-PCR was calculated by a Pearson correlation coefficient.

### 2.9. Measurement of Chlorophyll and Soluble Protein Concentrations in Leaf Tissues

The chlorophyll content and the absorbance of 0.1 g flag leaf at 645 and 663 nm in four periods were measured by a spectrophotometer. Refer to the Plant Chlorophyll Content Assay Kit (Boxbio AKPL003M, Beijing, China) for detailed experimental procedures and Chla and Chlb content calculations. The soluble protein content was determined using Coomassie brilliant blue [39].

## 3. Results

### 3.1. Statistics of Sequencing and Mapping Data

With Q30 percentages (sequencing error rate 0.1%) over 92.15%, a total of 269,360,020 paired-end raw reads were generated for four libraries. More than 97% of the raw reads were determined as clean data after quality filtering from individual libraries, and 62.37–65.57% were mapped to a single transcript. Most of these reads (over 88.80%) were aligned to exons (Table 1).

In sum, 76,872 transcripts found in at least two libraries and more than three aligned reads were obtained, of which 11,771 were novel. The raw data were submitted to the Sequence Read Archive (SRA) at the National Center for Biotechnology Information (accession number: SRP067916).

### 3.2. DEG Identification in Different Periods

Based on the cutoff values (|log2[foldchange]| > 1, *p* < 0.005), 4887 DEGs were identified among any two time points. At 0–15 DAA, 15–25 DAA, and 25–30 DAA, most of the DEGs (90.96%, 75.30%, and 77.04%) did not show significant changes in their expressions, respectively. The biggest shift of upregulated DEGs took place in the interval of 15–25 DAA (820) compared to other intervals, while downregulated DEGs continuously increased until the interval of 25–30 DAA (666). Only 29 DEGs were shared by all three intervals, and common genes between 0–15 DAA and either 15–25 DAA (92) or 25–30 DAA (67) were less than that between the latter two intervals (446). Around 55.08% of DEGs only showed differential expressions during the periods across one or two-time points (e.g., 0–25 DAA, 25–30 DAA, and 0–30 DAA). The DEGs in each interval are shown in Appendix A.

### 3.3. Functional Annotation of DEGs

As shown in Figure 1A, biological processes, including cell morphogenesis (GO:0000902), transport (GO:0006810), abiotic stimulus-response (GO:0009628), and phosphorus metabolism (GO:0006793), were enriched in the first 15 DAA. Expressions of most DEGs that related to the first three terms were upregulated, whereas DEGs involved in phosphorus metabolism, especially protein phosphorylation (GO:0006468), were downregulated. Carbohydrate (GO:0005975), lipid (GG:0006629), and amino acid metabolisms (GO:0006520) were active processes after anthesis. Upregulated DEGs (58) associated with lipid metabolism became remarkably enriched compared to the downregulated DEGs (19) at 15–25 DAA (Figure 1B). Compared with results during 15–25 DAA, downregulated DEGs for both carbohydrate and amino acid metabolisms were more significantly enriched than the upregulated DEGs at 25–30 DAA (Figure 1C). In contrast to the increase at 15–25 DAA, transcripts of all DEGs involved in the tricarboxylic acid (TCA) cycle (GO:0006099) and the coding of phosphofructokinase (PFK) (GO:0005945) were exclusively dropped at 25–30 DAA. Additionally, significant changes in chloroplast-related DEGs were common traits at both 15–25 DAA and 25–30 DAA (Figure 1).

The DEGs with reduced expression levels progressively increased until almost no upregulated DEGs were noted at 25–30 DAA, especially the genes that encoded functional units of photosystem I (PSI) (GO:0009523) and photosystem II (PSII) (GO:0009522) (Appendix A). The upregulated DEGs, including a transmembrane transporter (GO:0055085) and an amino acid transporter (GO:0006865), were enriched at 15–25 DAA and 25–30 DAA, respectively. In addition, cysteine-type peptidase DEGs and antioxidant-encoded DEGs were specifically enriched at 25–30 DAA.

### 3.4. KEGG Enrichment Analysis of DEGs

More KEGG pathways were significantly enriched for both up- and downregulated DEGs at 15–25 DAA (13/7) and 25–30 DAA (15/11) than at 0–15 DAA (5/4). Two pathways, glyoxylate and dicarboxylate metabolism and carbon fixation in photosynthetic organisms, were vigorous in all three sequential periods. Additionally, more pathways were shared by the 15–25 DAA and 25–30 DAA (3/5) intervals (Appendix A). All DEGs were sorted into 26 groups based on their expression dynamics, and 12 of these are shown in Figure 2. Although an overall downward trend was observed for major photosynthesis-related genes, minor distinctions in expressions were still detected among DEGs that individually related to carbon fixation in photosynthetic organisms (Figure 2B), carotenoid metabolism (Figure 2C,D), photosystem (Figure 2D), and chlorophyll metabolism (Figure 2C,J). Notably, only the transcripts of Rubisco-coded genes kept dramatically reducing after flowering (Figure 2A). DEGs that participated in carbohydrate, lipid, and protein metabolisms showed various expression patterns. For example, the average transcript amounts of major DEGs involved in glyoxylate and dicarboxylate metabolism and the pentose phosphate pathway dropped from 15 DAA and 25 DAA, respectively (Figure 2B,D).

Transcript amounts of DEGs for lipid metabolism reached their maximum at 25 DAA (Figure 2F,G) or 30 DAA (Figure 2K). Moreover, the genes associated with the metabolism of multiple amino acids were generally upregulated after 25 DAA (Figure 2F,H). As a crucial cofactor for metal element transport, DEGs for nicotinamide metabolism showed globally significant transcript accumulations at 25–30 DAA and 15–30 DAA (Figure 2H,K). The same trends were also observed for DEGs involved in the regulation of autophagy (Figure 2H,K).

### 3.5. Identification of SAGs

Over one-third of the DEGs were identified as putative SAGs (1780/4887) in this study, which was close homologs of SAGs previously found in 14 species. As expected, a large number of genes played roles in multiple substance metabolisms (30.24%), including the lipid, carbohydrate, amino acid, protein, nucleic acid, chlorophyll, and secondary metabolites. In addition, other SAGs were also involved in transcription regulation (5.22%), redox regulation (6.35%), transport (7.42%), and phytohormone and signal transduction (8.66%), among others. All information relating to putative SAGs is shown in Appendix A.

### 3.6. Transporter-Encoded DEGs

A total of 273 putative transporter-encoded DEGs were confirmed, and they were involved in the inter- or intracellular trafficking of a range of substances, including phytohormones, carbohydrates, amino acids, and minerals. Followed by carbohydrate transporter genes (16.48%), metal translocator genes were the most abundant group comprising over one-third of the transporter-coded DEGs (23.08%) (Appendix A).

According to the expression patterns, the transcripts of amino acid permease 3 (*AAP3*), bidirectional amino acid transporter 1 (*BAT1*), cationic amino acid transporter 2 (*CAT2*), and part of proline transporter 1 (*ProT1*) kept rising after anthesis, whereas that growth for most lysine histidine transporters *(LHTs*) started at 15 DAA (Figure 3A–D). The expressions were discrepant among carbohydrate translocator DEGs to some extent. Apart from an over five-fold (5.17) consistent transcript increase for one glucose-6-phosphate/phosphate translocator (*GPT*) at 15–30 DAA, others were only upregulated at 15–25 DAA (Figure 3E). Unlike various trends for other polyol transporters (*PLT*s), the upregulation of most *PLT5*s was observed at 15–30 DAA (Figure 3F). Followed by a slight increase (1.01–1.14-fold), the expression of a tonoplast sugar-efflux transporter, early responsive to dehydration 6-like 4 (*ERD6-like4*), was remarkably promoted before 25 DAA (2.71–3.44-fold) (Figure 3G). DEGs that code aquaporins showed significant transcription enhancements (2.35–4.28-fold) at 0–15 DAA and complex change trends after that (Figure 3H). Multiple metal transporter genes were differentially regulated. Two copper transporter genes, copper chaperone (*CCH*) and partial antioxidant 1 (*ATX1*), were upregulated at 15–30 DAA (Figure 3I), whereas the magnesium transport genes, mitochondrial RNA splicing 2-A (*MRS2-A*) and nonimprinted in Prader-Willi/Angelman syndrome region protein 2 (*NIPA2*), were downregulated during this period (Figure 3J).

About 34.92% of the metal transporter-encoded DEGs were confirmed as potassium-related, and they showed diversity in their expression trends (Figure 3K–N). Iron and zinc transporter genes were also differentially expressed. With a flat expression at 0–15 DAA (<1.5-fold), ferric reductase oxidase 7 (*FRO7*) transcripts showed dramatic drops afterward (Figure 3O). DEGs that were homologous with two heavy metal transporters (*HMA*s) and two vacuolar iron transporters (*VIT*s) were identified. The expression pattern of *HAM1* reversed that of *FRO7*, and *VIT1* showed opposite changes during the first two periods, namely a steep rise at 0–15 DAA, followed by a sharp drop at 15–25 DAA (Figure 3Q). Four different yellow stripe-likes (*YSL*s) were DEGs, and most of them were upregulated after flowering (except one *YSL12* and one *YSL14*) (Figure 3S).

### 3.7. Transcription Factors

There were 164 differentially expressed TFs that belonged to 27 different families, and NAC (19.52%) constituted the largest TF family (Appendix A). All TFs were divided into 14 groups using a hierarchical cluster analysis (Figure 4). Most TFs (54.88%) were concentrated in three clusters, namely clusters 11 (32), 12 (26), and 14 (32) (Figure 4K–N). Among these genes, TFs of clusters 11 and 12 showed an average of 11.44-fold upregulated expressions and 5.34-fold downregulated expressions after a flat change at 0–15 DAA (1.16/1.02-fold), respectively. Over half of the *NAC* genes (18) belonged to cluster 11, including the homologs of *AeCUC2*, *AeNAC29*, *TuNAC8*, and *TaNAC6*. Additionally, the closest ortholog of auxin response factor 18 (*OsARF18*) was in this group. Major TFs with putative roles in basic biological activities were in cluster 12, such as the homolog genes of golden-like 2 (*TuGLK*2), agamous-like 10 (*TaAGL10*), CO-like 4 (*OsCOL4*), and rice RNA polymerase sigma factor genes. Compared with the nearly 4-fold rise in cluster 11 (3.95-fold) at 25–30 DAA, the overall expression change of the DEGs in cluster 14 was insignificant (1.30-fold). The homologs of eight *NACs* (*AeCUC2* and *AeNAC29*), IAA-leucine resistant 3 (*OsILR3*), *OsMYB1R1*, and RBA-related 1 (*TaRBR1*) were in this class. Notably, only three previously characterized *NAC* genes (*AeNAM-D1*, *TtNAM-A1*, and *TtNAM-B2*), which control the reserve substance redistribution and flag leaf senescence in wheat, were found in cluster 13. After a rapid rise in the transcript levels (7.0–15.99-fold), their increases became slower after 15 DAA (1.08–2.55-fold) (Figure 4M). Only 6.10% of DEGs were defined as *WRKY* genes, and these genes mainly showed two general types of expression trends (Figure 4B–D,F,H). At 0–15 DAA, the transcripts of all *WRKY* genes were dramatically reduced (4.81–8.99-fold), except for a slight decline (1.42-fold) for one *TuWRKY6* (*OsWRKY1*) homolog. After this point, four *WRKY* genes (including *TaWRKY74*) in cluster 8 continued to be modestly downregulated until 30 DAA (<4.50-fold). However, the expressions of *TaWRKY53*, *TaWRKY14*, and *TaWRKY2* in cluster 2 (>8.54-fold) were promoted more sharply than those of *TaYWRK27* in cluster 4 (4.71-fold) and *TuWRKY41* in cluster 3 (1.97-fold) (Figure 4).

### 3.8. Phytohormone-Related DEGs

A local blast facility against an Arabidopsis phytohormone database was applied to identify 124 putative phytohormone-associated DEGs, some of which were simultaneously implicated in multiple phytohormones. These genes were likely involved in the synthesis, transport, and signal transduction of seven phytohormones, including abscisic acid (ABA), jasmonic acid (JA), ethylene (ET), cytokinin (CTK), brassinosteroid (BR), salicylic acid (SA), and auxin (IAA). IAA (44), JA (33), and ABA (28) were the three phytohormones with the most abundant DEGs (Appendix A).

The expression trends of some core components for these processes are shown in Figure 5. Unlike the transcript increase of pyracbactin resistance-like 8 (*PYL8*) after 15 DAA, other ABA receptors, namely genomes uncoupled 5 (*GUN5*) and *PYL4*, declined after 15 DAA or 0 DAA, respectively (Figure 5A). Moreover, type 2C protein phosphatase (*PP2C*) family genes, the negative signaling regulator of ABA, were all upregulated during 0–25 DAA, while the positive regulator respiratory burst oxidase (*RBOH*) was only upregulated at 15–25 DAA. Multiple IAA transporter genes were also differentially expressed. Genes encoding influx transporter like-AUX1 3 (LAX3) and efflux transporter aminopeptidase (M1APM1) were down- and upregulated during 0–30 DAA, respectively. Efflux transporter P-glycoprotein 11 (PGP11) was upregulated only until 25 DAA (Figure 5B). Evident expression correlations among DEGs for JA biosynthesis were not observed. For instance, the expression patterns of omega-3 fatty acid desaturase 7 (*FAD7*) and phospholipase D (*PLD*) at 15–30 DAA reversed that of another *FAD7*, 12-oxophytodienoate reductase 1 (*OPR1*), as well as 3-ketoacyl-CoA thiolase 2 (*KAT2*) (Figure 5F). A difference between two core components of JA signal transduction, namely coronatine insensitive1 (COI1) and TIFY10A/B, was detected only at 25–30 DAA (Figure 5F). Information about the DEGs associated with other phytohormones was sparse and complex. As with the signaling repressors of BR, the expression dynamics for growth regulating factor 3 (*GRF3*) and shaggy-related protein kinase (*ASK41*) were diverse (Figure 5C). Despite disparate functions of histidine kinase 3 (AHK3) and the two-component response regulator ARR12 (ARR12) in the CTK signal pathway, their encoded genes were similarly upregulated (Figure 5D). The expression of ETHYLENE INSENSITIVE3 (*EIN3*), a positive regulator in the ET-signaling system, was partially coordinated with that of the ET synthesis gene s-adenosylmethionine synthase (*SAM2*) (Figure 5E).

### 3.9. Verification Analysis of qRT-PCR to RNA-seq

To validate the results of RNA-seq, we selected 12 genes from different pathways of SAGs to perform a quantitative real-time PCR (qRT-PCR) analysis. These results showed a similar trend to our RNA-seq results, giving further credence to our sequencing findings (Appendix A). For instance, among the transporter-related genes, *Traes_2AS_9DE16F020* and *Traes_2AS_9DE16F020* had a trend of decreasing first and then increasing, and both had the highest expression at 30 DAA. The expression of four TF genes, *Traes_7DS_F5A240B02*, *Traes_2AL_8A23618BA*, *Traes_1BS_EF67E5A24*, and *Traes_1AS_F3EAEC435* increased from post-anthesis to 30 DAA. *Traes_5BL_62D9B877B*, phytohormone-related genes, had the same trend. Chlorophyll synthesis-related genes *Traes_6DL_0FF72D765* and *Traes_2DL_30F23E577* gradually decreased from 0 DAA or 15 DAA to 30 DAA. The expression of *Traes_2BL_98439EA10* and *Traes_2DS_AE6E354A9* in 0–15 DAA gradually increased but then gradually decreased. The correlation heatmap showed a good correlation between RNA-seq and qRT-PCR, which indicated the accuracy of our sequencing data (Appendix A).

## 4. Discussion

The flag leaf is the last leaf that grows before the emergence of the wheat spike, and it is an important sign of wheat entering the booting stage. Under suitable growth conditions, 80% of the dry matter of wheat grains is accumulated after anthesis, and the contribution rate of “functional leaves”, i.e., flag leaves, to wheat grain yield can be as high as one-third. In addition to photosynthetic compounds, the senescence of flag leaves also provides some basic materials for the vigorous growth of grains. Previous studies have shown that the advance in senescence of vegetative tissues will be conducive to the accumulation of more protein and essential trace elements in grains [13]. The wheat flag leaf serves as the major N source that provides a large amount of assimilates for grains at the post-anthesis stage and serve as a model tissue to study leaf senescence and N remobilization [16]. Therefore, the senescence of wheat flag leaf is an important factor that affects the main agronomic traits, including yield and nutritional quality. The study on the senescence mechanism of flag leaf will help to improve these agronomic characteristics.

The cluster analysis in the present study showed that expression trends of major DEGs were different before and after 15 DAA (Appendix A). Compared with 0–15 DAA, more metabolic pathways were enriched at 15–25 DAA and 25–30 DAA. Among them, more genes involved in the metabolism of fat, amino acids, and carbohydrates are more active in 15–25 DAA. Additionally, there was more similarity between these two stages than at 0–15 DAA. This suggested that 15 DAA was a crucial turning point for the metabolism in the flag leaf, and it reflected a response of metabolic changes in the flag leaf to the demand of the grain as well. Grain development consists of two phases after anthesis: cell division and grain filling. In the first stage, the kernel length and width are established until approximately 14 DAA [40], but not much dry matter is accumulated. This corresponds with the slow-rate dry matter accumulation in the grain before 12 DAA [41]. The downregulated genes increased continuously after anthesis and reached the maximum at 25–30 DAA. This indicates that, at the molecular level, 15–25 DAA may be a transitional period for the metabolic degree of Chinese Spring flag leaf cells. At this time, there may be catabolism and anabolism in the cell, and anabolism is stronger than catabolism. In addition, multiple metabolisms at 25–30 DAA became less active than those at 15–25 DAA, likely due to the progressive function loss of chloroplasts after 15 DAA. In the mature grains, 66.7% of the total N was remobilized from the pre-anthesis accumulation in the biomass, while the remaining 33.3% was derived from the N taken up during post-anthesis. From anthesis to 2 weeks after the anthesis stage, the flag leaf remobilized 3.67 mg of N outwards, and the ear remobilized 3.87 mg of N inwards from the pre-anthesis accumulation in each plant [42].

The DEGs related to photosynthesis also changed significantly from 15 DAA. Whether it is the functional components of the photosynthetic system or other related genes involved in photosynthesis, the number and types of 25–30 DAA downregulated genes were significantly increased compared with 15–25 DAA. Differently, only enriched downregulated DEGs encoding components of the photosynthetic system were observed at 25–30 DAA. Although leaf yellowing is a major visible symptom of senescence, the main factor that causes photosynthesis inactivation during this process remains under discussion. In wheat, parallel losses of photosynthetic activity and chlorophyll content have been reported [43], whereas others found no obvious correlation existed [44]. With a flat expression at 0–15 DAA, DEGs that encode enzymes such as pheophorbide an oxygenase (PaO) and chlorophyll (ide) b reductases (CBR) for chlorophyll breakdown were significantly upregulated after 15 DAA, while DEGs involved in chlorophyll synthesis were downregulated. Therefore, the rapid reduction in the chlorophyll content may start at 15 DAA or later (Appendix A). Additionally, Grover et al. (1993) [45] (pp. 225–255) found that chloroplast function loss was associated with the decline in photochemical activities of PSI and PSII due to the chloroplast disintegration in wheat. In the present study, dramatic changes in the DEG coding functional photosystem units were observed after 15 DAA. Furthermore, more DEGs related to PSII (11) showed reduced expressions than those related to PSI (6) at 15–25 DAA, whereas, after, a larger increase in the number of PSI-encoded genes (18) was detected compared to PSII (18) genes. Several studies have indicated diverse physiological changes between PSI and PSII during leaf senescence. For example, PSII was more susceptible to senescence than PSI [46,47]. It has also been reported that the decline of PSII activity preceded that of PSI activity in wheat leaves during heat-promoted senescence [10]. Despite the rate-limiting enzyme for photosynthesis in the C3 plant [48], the role of Rubisco in photosynthetic activity decline has been debated [45,49,50,51,52]. Interestingly, Rubisco-encoded genes were the only DEGs that showed consistently and significantly decreased expressions after anthesis in the present study. Considering a normally rapid drop of net photosynthesis after 15 DAA [53], the overall results suggested that a chlorophyll decrease or photosystem breakdown had major effects on photosynthetic inactivation, rather than the reduced amounts of Rubisco (Appendix A). Zhou et al. (2018) [42] also reported that Rubisco might play a critical role in N deposition.

Regardless of the photoassimilate or storage resource, transporters are undoubtedly indispensable for their distributions of grain. As one of the major long-distance transport forms of organic nitrogen, the translocation of amino acids depends on various membrane-integral transporters in plants [54]. The expression trend of each transporter was complex, and the expression of genes involved in the same function was also different. AAPs were involved in the phloem trafficking of amino acids supplied to the sink [55,56]. The rapid transcript rise of AAPs in the flag leaf 15 DAA may also reflect their response to the growing demand for grain. Furthermore, magnesium (Mg) ion was not only the central atom of the chlorophyll molecule but also specifically affected the half-lives of chloroplast RNA [57]. Three Mg transporter DEGs were remarkably downregulated during 15–30 DAA, which likely resulted from chloroplast degradation. TaFRO1 and TaFRO2 (ferric reduction oxidation) encode oxidoreductases localized to the chloroplast membrane. This enzyme was responsible for the reduction of Fe^3+^ to Fe^2+^ before the transport of Fe^3+^ to the chloroplast cavity. Consistent with previous studies, the expression of *TaFRO1* and some *TaFRO2* genes decreased at 15–30 DAA [27]. Both HMA1 and FRO7 reside in chloroplast membranes; however, they are functionally different. HMA1 is a member of the Zn/Co/Cd/Pb-transporting HMA family, and its close homolog in rice is shown to be able to export metals across the chloroplast [58]. In the present study, expression changes implied that the efflux of Zn/Fe from chloroplasts was more active than their influx at 15–30 DAA. The vacuole was a crucial organelle to buffer the concentration of metal in the cell. A few tonoplast-integrated transporter genes were differentially expressed. A previous study suggested a potential contribution of *OsVIT1* to the recycling of Zn/Fe during senescence [59]. At 0–30 DAA, the expression level of *TaVIT1* decreased by about 2.4–4.5-fold, while *Ta VIT2* decreased from 15 DAA. Consistent with previous research, only NRAMP2, a member of the NRAMP family, was identified as the DEG during grain filling [60]. Moreover, the continuous upregulation at 15–30 DAA confirmed that this gene facilitated the iron efflux from the vacuole to satisfy the material demand during grain development.

Many identified NAC and WRKY TFs have been associated with leaf senescence in Arabidopsis, barley, and wheat [15,21,61,62]. Over half the *NAC* genes identified in this study showed a correlation with senescence based on their expression patterns, including *NAC21/22*, *NAC29*, *NAC8*, *NAC7*, and three other characterized *NAM* genes. Although less direct evidence has been established, several studies ultimately proved the involvement of their homologs in aging or stress responses. For example, it was shown that ectopic expression of *TaSNAC11-4B* in Arabidopsis promotes ROS accumulation and significantly accelerated age-dependent, as well as drought- and ABA-induced, leaf senescence. *TaSNAC11-4B* was shown to positively regulate the expressions of *AtrbohD* and *AtrbohF*, which encode catalytic subunits of ROS-producing NADPH oxidase [21]. *TaNAC21/22* negatively regulates the resistance of wheat to stripe rust [63], while *TaNAC8*, *TaNAC29*, and *TaNAC7* are also related to the defense responses [64,65,66]. *NAM-A1*, *NAM-B2*, and *NAM-D1* were expressed in a generally similar trend to that established in a previous study in tetraploid wheat [13]. The present study verified the dramatic increase in their transcript amounts of those regulatory genes exclusively at 0–15 DAA in early senescence.

Research conducted by Lu et al. (2022) [67] showed that the yield of a premature senescence mutant (GSm) was considerably lower than that of the wild type (WT). Many physiological indexes are lower than those of the WT, except malondialdehyde. The transcriptomic analysis indicated that blockades of chlorophyll and carotenoid biosynthesis accelerate the degradation of chlorophyll and diminish the photosynthetic capacity in mutant leaves and that brassinolide may facilitate chlorophyll breakdown and, consequently, accelerate leaf senescence. This research also found that *NAC* genes positively regulate the senescence process. Compared with *NAC* genes, expressions of the *WRKY* and *MYB* genes are induced earlier in the mutant, possibly due to increased levels of reactive oxygen species and plant hormones (e.g., brassinolide, salicylic acid, and jasmonic acid), thereby accelerating leaf senescence. Furthermore, this research ascertained that the antioxidant system plays a role in minimizing oxidative damage in the mutant [67]. Mostly, the genes detected in the carotenoid biosynthetic pathway initiated by ABA (*PYR/PYL*, *PP2C*, *SnRK2,* and *ABF*) expressed higher in either no or low N than high N. ABA is known to promote senescence in the plant by the stomatal closure, which restricts cellular growth and induces SAGs, *non-yellow coloring 1(NYCI)*, *stay green (SGR)*, *PHEOPHYTINASE (PPH),* and *pheophorbide a**n oxygenase (PAO)* gene expressions. The higher expression of carotenoid-related genes may be attributed to the chlorophyll degradation observed in the N-limited conditions (no and low N) compared to the high N [16].

In the past 20 years, *WRKY* TFs have been broadly investigated in plants, and they were extensively proven to be involved in numerous physiological processes, including leaf senescence. For example, *WRKY14*, *WRKY2*, *WRKY53*, WRKY13, *WRKY40*, and *WRKY42* are putative SAGs, as their close homologs are positive regulators of JA- and ET-mediated signaling [23,24,25,68]. Likewise, in the present study, these genes were upregulated as the flag leaf senesced. Inverse expression patterns between phytochrome-interacting factor 3 (PIF3), PIF5, and GLK2 confirmed the nature of their interactive relationships. PIFs are central mediators in phytochrome signaling [69], and a study on Arabidopsis proved that certain ones (PIF3, PIF4, and PIF5) are required to restrain photomorphogenic development by repressing chloroplast activity maintainer gene GLK2 [70]. Accumulated evidence implies a more complicated network is involved in the regulation of leaf senescence. Although it has been suggested that CTK and IAA have positive roles in plant development, many studies have found these phytohormones also have functions in leaf senescence [71]. CTK-induced changes in nitrogen remobilization and chloroplast ultrastructure in wheat retain the sink activity of the older leaves by inhibiting amino acid and sugar export to the phloem and stimulating assimilate accumulation in the chloroplasts of the older leaves [72]. The JA, ABA, SA, and BR-related genes promoted leaf senescence in low N, whereas the IAA, GA, and CTK genes inhibited leaf senescence in hight N [16]. In line with a similar study in cotton, IAA polar transporter genes showed active changes as well [73]. Furthermore, the efflux of IAA was more vigorous than its influx in the present study. This result again verified the changes in IAA transport, rather than the endogenous IAA level itself, which could confer growth plasticity. Additionally, many DEGs were simultaneously modulated by multiple phytohormones, which demonstrated their extensive and complicated crosstalk during our sampling period. To date, few core genes have been identified for signaling pathways. This restricts a clear picture of the role of each phytohormone, and their relationships in the regulation of senescence were explored based on the expressions of related genes.

## 5. Conclusions

By analyzing the changes of the flag leaf transcriptome of Chinese Spring wheat at 0DAA, 15DAA, 25DAA, and 30DAA, the differential expression of senescence-related genes was investigated, which is helpful to understanding the relevant pathways and regulation of wheat senescence. We confirmed that different periods lead to the rearrangement of glyoxylic acid, dicarboxylic acid metabolism, carotenoid metabolism, photosystem, chlorophyll metabolism, carbohydrate, lipids, and protein metabolism. Some transporters, such as amino acid permease 3, bidirectional amino acid transporter 1 (*BAT1*), cationic amino acid transporter 2 (*CAT2*), part of proline transporter 1 (*ProT1*), lysine histidine transporters (*LHTs*), metal transporter genes, etc., were identified to be associated with senescence. Some NAC and WRKY family transcription factors such as *T**aWRKY53*, *T**aWRKY14*, and *T**aWRKY2*, and senescence genes of ABA, JA, ET, CTK, BR, SA, and IAA related to plant hormones were identified. The results provided in this paper enrich the senescence gene resources and further provide a basis for improving the wheat yield and quality at the same time. Future research may clone some of these genes and further analyze their functions and regulatory mechanisms.

## Figures and Tables

**Figure 1 plants-11-02593-f001:**
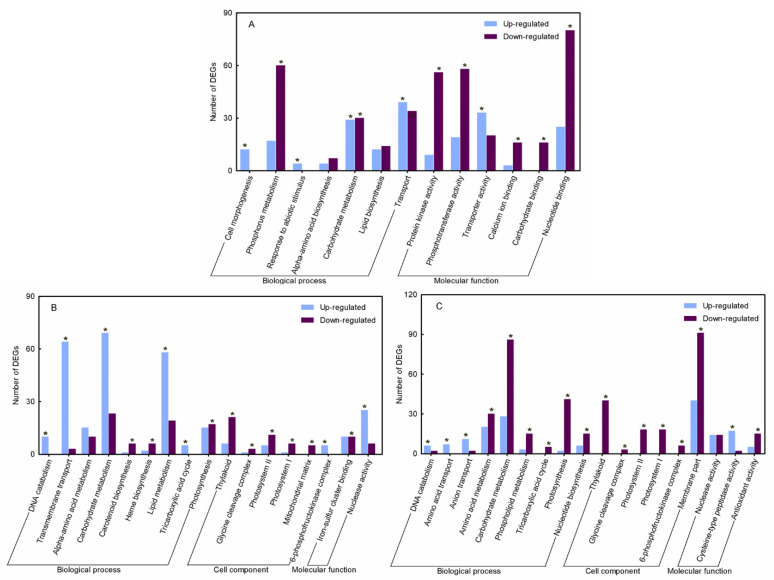
Enrichment analysis of GO terms in three sequential periods. (**A**) 0–15 DAA. (**B**) 15–25 DAA. (**C**) 25–30 DAA. “*”: Significantly enriched GO terms, *p* < 0.05.

**Figure 2 plants-11-02593-f002:**
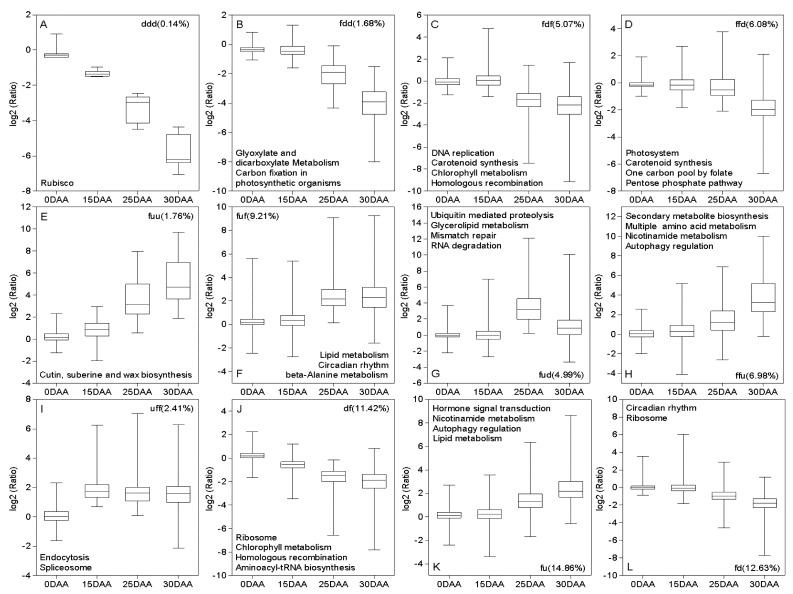
Classification of DEGs based on their expression dynamics. Only 12 groups (**A**–**L**) with enriched KEGG pathways are shown. “f” refers to the flat expression; “d” and “u” represent significant down- and up-regulation, respectively. The number in parentheses is the percentage of DEGs in each class. (**A**–**I**) three letters, for example, “ddd” indicate down-regulation at 0–15 DAA, 15–25 DAA and 25–30 DAA. (**J**) “df” indicates DEGs with downregulation expressions at 0–25 DAA and flat expressions at 25–30 DAA. (**K**,**L**) “fu” and “fd” indicate DEGs with flat expressions at 0–15 DAA and differential expressions at 15–30 DAA. The median (horizontal lines), 25th and 75th percentiles (boxes), and expression ranges (whiskers) are shown for each group.

**Figure 3 plants-11-02593-f003:**
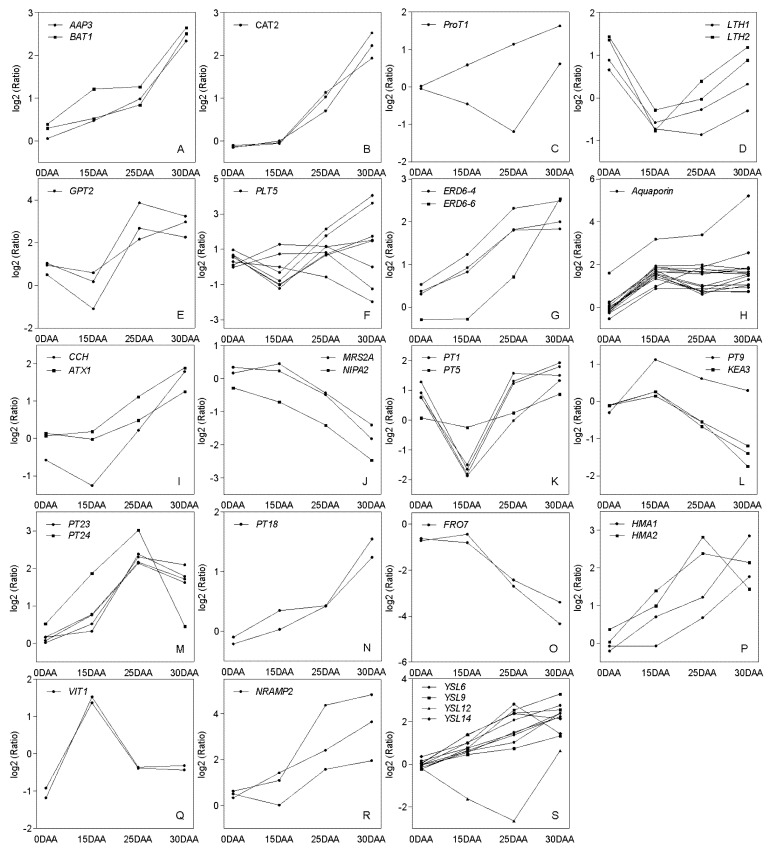
Expression trends of differentially expressed transporter genes. (**A**–**D**) Amino acid transporter genes. (**E**–**G**) Carbohydrate transporter genes. (**H**) Aquaporin transporter genes. (**I**) Cu transporter genes. (**J**) Mg transporter genes. (**K**–**N**) K transporter genes. (**O**–**S**) Fe and Zn transporter genes.

**Figure 4 plants-11-02593-f004:**
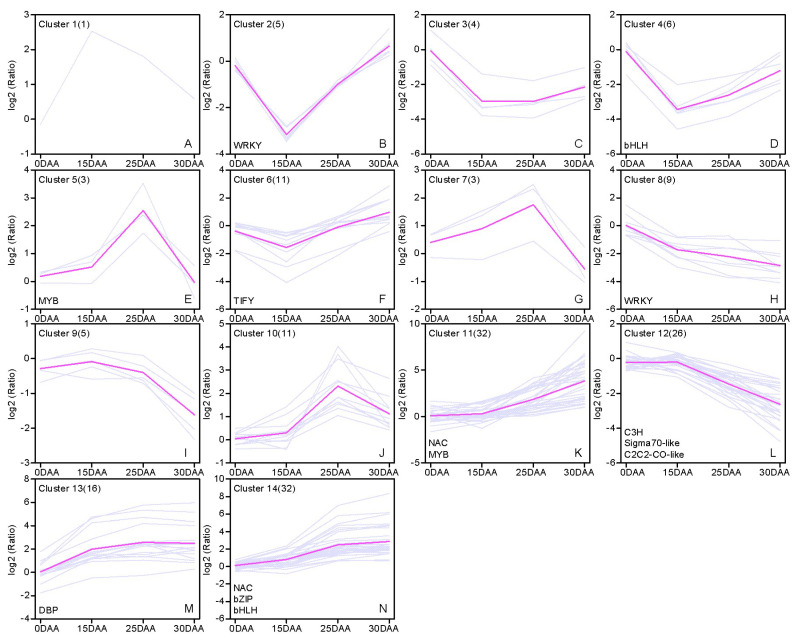
Cluster analysis of differentially expressed transcription factor genes. (**A**–**N**) The expression pattern of transcription factors of different families in the 0–15 DAA, 15–25 DAA, and 25–30 DAA period. Different colorful lines show the overall trend of genes expression in three periods. TF families are noted in the cluster where most of their members belong. The figure in parentheses indicates the gene number in the group.

**Figure 5 plants-11-02593-f005:**
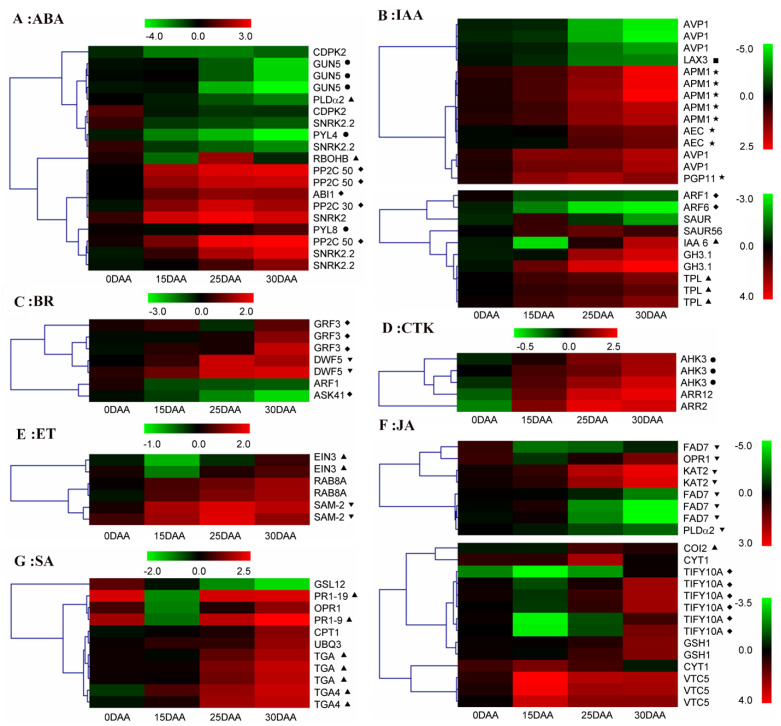
Cluster analysis of phytohormone-related DEGs. (**A**) ABA. (**B**) IAA. (**C**) BR. (**D**) CTK. (**E**) ET. (**F**) JA. (**G**) SA. “●”: Phytohormone receptor, “▲”: Positive regulator, “◆”: Negative regulator, “■”: Influx transporter, “★”: Efflux transporter, and “▼”: Synthesis gene.

**Table 1 plants-11-02593-t001:** Summary of transcriptome sequencing data.

Sample Name	0DAA	15DAA	25DAA	30DAA
Raw reads	66,015,110	64,686,904	65,524,154	73,133,852
Clean reads	64,279,134	62,887,574	63,658,066	71,300,896
Q30 (%)	92.23	92.15	92.2	92.44
Total mapped	47,615,252 (74.08%)	46,028,993 (73.19%)	46,574,821 (73.16%)	49,556,485 (69.50%)
Uniquely mapped	42,096,708 (65.49%)	40,727,648 (64.76%)	41,740,379 (65.57%)	44,467,404 (62.37%)
Exon mapped (%)	90.1	89.9	89.7	88.8

## Data Availability

The raw data were submitted to the Sequence Read Archive (SRA) at the National Center for Biotechnology Information (accession number: SRP067916).

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
