# Peer review of "Transcriptome Analysis of Early Senescence in the Post-Anthesis Flag Leaf of Wheat (Triticum aestivum L.)"

_plants, 2022, doi:10.3390/plants11192593_

Round 1

Reviewer 1 Report

The paper plants-1851706 is in the focus of the journal and bring some information about the senescence process in flag leaves of wheat, although most of the conclusions have been described before. The main conclusion is that leaf transcriptome is highly regulated, and that the progression of early senescence is regulated- among others- by NAC and WRKY TFs, and phytohormones, which is well-known. Nevertheless, the study claims to identify and characterize a large set of genes and metabolic discrepancies, but some physiological confirmations would benefit to support this (i.e. chlorophyll, amino acids or soluble protein determinations).

Some sections are difficult to follow and some ideas are disconnected. The writing style needs some attention and revamp. For example, authors mention the importance of leaf senescence in nutrients remobilization in L43-53 and then again in L92-95. The authors give 40 lines to introduce WRKY, NAC and other TFs but the importance of C and N remobilization in this stage is barely mentioned despite is crucial for the new organs formation.

In the discussion, the interactions between photosynthesis, phytohormones, carbon and nitrogen remobilization and the senescence process should be deeply described. 

Although literature is updated, there are missing some metabolomic and transcriptomic studies that would enrich the discussion of this work. In addition, some mistakes have been observed in the reference list.

Relevant information is missing in the material and methods section. 

Conclusions are too generic and there is no clear take home message.

I think that the manuscript needs a major revision and that the authors considered those comments and included the information in the paper.

Specific comments:

Please homogenize font and size type of the whole manuscript.

L103- hypothesis and objectives need attention.

L105. Gregesrsen et al (2008) is a review. Did authors want to mention Gregersen and Holm Plant Biotechnology Journal (2007) 5, pp. 192–206?

L113- More details about the growing conditions are needed. Was wheat grown in pots or in plots? How many rows and plants per row had the plots? How many plots were stablished? Were plants watered and fertilized? How was the temperature and water status of the crop? Which fertilize was used (if any)? When was the crop planted and under which environmental conditions grew?

How were leaves collected and processed? Were they collected and placed in liquid nitrogen? Were the flag leaves collected from different plants?

All that information is extremely meaningful for readers in order to compare and replicate the work.

L168- Was the validation analysis performed in different leaves that the used in the RNAseq? Although it is mentioned that the analysis was made in only 4 genes, there are 9 genes in Table S1 and Figure S4. In addition, Traes_5AL_F96AEDDB8 and Traes_5BL_62D9B877B are reported as different genes that codify for the same enzyme (EC 1.2.1.3). I suggest to include a WRKY TFs but also other genes that have a downregulation profile (Rubisco, chlorophyll synthesis, amino acids…) to increase the number of genes used for the RNA-seq validation.

L191- Principal component analysis of the RNA-seq data from the 10 replicates from each treatment would allow a better visualization of the relationship between the samples. 

L371- Validation of DEGs by qRT-PCR deserves its own subheading. Also, how good/bad is the correlation between the RNA-seq and qRT-PCR? A person correlation analysis is required to confirm the expression levels found in the RNA-seq.

L418-421- I suggest that authors confirm it by determining chlorophyll content.

L429. Are there more recent works?

L435-439. Similar to chlorophyll contents, the physiological confirmation by determining soluble protein contents is necessary to validate RNA-seq results.

Author Response

Thansk so much for your time, critical reading of the manuscript, and constructive suggestions, which significantly improved the quality of our work. The manuscript is now thoroughly revised, and a brief response to the review’s comments is provided below.

Responds to the comments:

  1. Please homogenize font and size type of the whole manuscript.

Response: We made the corrections.

  1. Gregesrsen et al (2008) is a review. Did authors want to mention Gregersen and Holm Plant Biotechnology Journal (2007) 5, pp. 192–206?

Response: Yes, we want to cite “Gregersen and Holm Plant Biotechnology Journal (2007) 5, pp. 192–206”, and made the changes in the mansucript.

  1. L168- Was the validation analysis performed in different leaves that the used in the RNAseq? Although it is mentioned that the analysis was made in only 4 genes, there are 9 genes in Table S1 and Figure S4. In addition, Traes_5AL_F96AEDDB8 and Traes_5BL_62D9B877B are reported as different genes that codify for the same enzyme (EC 1.2.1.3). I suggest to include a WRKY TFs but also other genes that have a downregulation profile (Rubisco, chlorophyll synthesis, amino acids…) to increase the number of genes used for the RNA-seq validation.

Response: The leaves used by qRT-PCR are the same as used forRNA-seq sequencing. Only 4 genes are ambiguous in our expression. We want to show that these 9 genes come from 4 different functional pathways and have been modified.

Due to Traes_5AL_F96AEDDB8 and Traes_5BL_62D9B877B are the same, we chose Traes_5BL_62D9B877B. In addition, we added the validation of WRKY TFs (Traes_1BS_EF67E5A24, Traes_1AS_F3EAEC435) and chlorophyll synthesis (Traes_6DL_0FF72D765.1,Traes_2DL_30F23E577.1) related genes.

The figure is placed in supplementary material II as Figure S4.

  1. L371- Validation of DEGs by qRT-PCR deserves its own subheading. Also, how good/bad is the correlation between the RNA-seq and qRT-PCR? A person correlation analysis is required to confirm the expression levels found in the RNA-seq.

Response: The correlation between aeq-RNA and qPCR was calculated by pearson-correlation coefficient. The figure is placed in supplementary material II as Figure S5.

  1. L418-421- I suggest that authors confirm it by determining chlorophyll content.

Response: We measured the chlorophyll content of flag leaves at four developmental stages to support this, and the figure is placed in supplementary material II as Figure S6.

  1. L429 (PSII was more susceptible to senescence than PSI). Are there more recent works?

Response: Mattila et al. (2021) approved this view.

Mattila H, Sotoudehnia P, Kuuslampi T. et al (2021) Singlet oxygen, flavonols and photoinhibition in green and senescing silver birch leaves. Trees 35:1267–1282 . https://doi.org/10.1007/s00468-021-02114-x

  1. L435-439. Similar to chlorophyll contents, the physiological confirmation by determining soluble protein contents is necessary to validate RNA-seq results

Response: We measured the soluble protein content of flag leaves at four developmental stages to support this, and the figure is placed in supplementary material II as Figure S6.

  1. For example, authors mention the importance of leaf senescence in nutrients remobilization in L43-53 and then again in L92-95. The authors give 40 lines to introduce WRKY, NAC and other TFs but the importance of C and N remobilization in this stage is barely mentioned despite is crucial for the new organs formation.If the authors can also expound upon some more N-assimilation and remobilization pathways that will be related to leaf senescence as well. 

Response: This part was modified in the text.

  1. Provide a clear methodology that must be clearly described.

Relevant information is missing in the material and methods section.

L113- More details about the growing conditions are needed. Was wheat grown in pots or in plots? How many rows and plants per row had the plots? How many plots were stablished? Were plants watered and fertilized? How was the temperature and water status of the crop? Which fertilize was used (if any)? When was the crop planted and under which environmental conditions grew? How were leaves collected and processed? Were they collected and placed in liquid nitrogen? Were the flag leaves collected from different plants?

Response: We added more details in the text according to the Reviewer’s suggestion.

  1. The discussion section is superfluous and must be robust and comprehensive.

Conclusions are too generic and there is no clear take home message.

In the discussion, the interactions between photosynthesis, phytohormones, carbon and nitrogen remobilization and the senescence process should be deeply described. 

Response: We added some results in the discussion

  1. Please homogenize font and size type of the whole manuscript.

Response: We made the corrections.

  1. Gregesrsen et al (2008) is a review. Did authors want to mention Gregersen and Holm Plant Biotechnology Journal (2007) 5, pp. 192–206?

Response: Yes, we want to cite “Gregersen and Holm Plant Biotechnology Journal (2007) 5, pp. 192–206”, and made the changes in the mansucript.

  1. L168- Was the validation analysis performed in different leaves that the used in the RNAseq? Although it is mentioned that the analysis was made in only 4 genes, there are 9 genes in Table S1 and Figure S4. In addition, Traes_5AL_F96AEDDB8 and Traes_5BL_62D9B877B are reported as different genes that codify for the same enzyme (EC 1.2.1.3). I suggest to include a WRKY TFs but also other genes that have a downregulation profile (Rubisco, chlorophyll synthesis, amino acids…) to increase the number of genes used for the RNA-seq validation.

Response: The leaves used by qRT-PCR are the same as used forRNA-seq sequencing. Only 4 genes are ambiguous in our expression. We want to show that these 9 genes come from 4 different functional pathways and have been modified.

Due to Traes_5AL_F96AEDDB8 and Traes_5BL_62D9B877B are the same, we chose Traes_5BL_62D9B877B. In addition, we added the validation of WRKY TFs (Traes_1BS_EF67E5A24, Traes_1AS_F3EAEC435) and chlorophyll synthesis (Traes_6DL_0FF72D765.1,Traes_2DL_30F23E577.1) related genes.

The figure is placed in supplementary material II as Figure S4.

  1. L371- Validation of DEGs by qRT-PCR deserves its own subheading. Also, how good/bad is the correlation between the RNA-seq and qRT-PCR? A person correlation analysis is required to confirm the expression levels found in the RNA-seq.

Response: The correlation between aeq-RNA and qPCR was calculated by pearson-correlation coefficient. The figure is placed in supplementary material II as Figure S5.

  1. L418-421- I suggest that authors confirm it by determining chlorophyll content.

Response: We measured the chlorophyll content of flag leaves at four developmental stages to support this, and the figure is placed in supplementary material II as Figure S6.

  1. L429 (PSII was more susceptible to senescence than PSI). Are there more recent works?

Response: Mattila et al. (2021) approved this view.

Mattila H, Sotoudehnia P, Kuuslampi T. et al (2021) Singlet oxygen, flavonols and photoinhibition in green and senescing silver birch leaves. Trees 35:1267–1282 . https://doi.org/10.1007/s00468-021-02114-x

  1. L435-439. Similar to chlorophyll contents, the physiological confirmation by determining soluble protein contents is necessary to validate RNA-seq results

Response: We measured the soluble protein content of flag leaves at four developmental stages to support this, and the figure is placed in supplementary material II as Figure S6.

  1. For example, authors mention the importance of leaf senescence in nutrients remobilization in L43-53 and then again in L92-95. The authors give 40 lines to introduce WRKY, NAC and other TFs but the importance of C and N remobilization in this stage is barely mentioned despite is crucial for the new organs formation.If the authors can also expound upon some more N-assimilation and remobilization pathways that will be related to leaf senescence as well. 

Response: This part was modified in the text.

Reviewer 2 Report

I believe the authors were able to address and present the objectives of their study in a clear and concise way. The pathways involved in senescence in itself is complex. Though the results and conclusions presented in this work is not novel per se, it is still vital information to corroborate previous and future studies to elucidate the mechanisms of senescence particularly in relation to grain yield. I understand this is not part of the current study, but it will be great if the authors can also expound upon some more N-assimilation and remobilization pathways that will be related to leaf senescence as well. 

Author Response

Thanks for your time, critical reading of the manuscript, and constructive suggestions, which significantly improved the quality of their work. The manuscript is now thoroughly revised.

Reviewer 3 Report

The authors used a single bread wheat cultivar and flag leaves were analysed during different post-anthesis phases at 0, 15, 25, and 30 days after anthesis. The main findings of the MS are, the identification of differentially expressed genes and their association with phytohormones (JA, ABA, IAA, ET, SA, BR, and CTK). Given leaf senescence process has already been studied in detail – some of the published articles are below, I don’t see any novel findings from this study except descriptive gene regulation analysis.

Transcriptome analysis of senescence in the flag leaf of wheat (Triticum aestivum L.)." Plant Biotechnology Journal 5, no. 1 (2007): 192-206.

Transcriptome profiling reveals major structural genes, transcription factors and biosynthetic pathways involved in leaf senescence and nitrogen remobilization in rainfed spring wheat under different nitrogen fertilization rates. Genomics114(2), 110271.

Identification of transcription factors regulating senescence in wheat through gene regulatory network modelling." Plant Physiology 180, no. 3 (2019): 1740-1755.

A WRKY transcription factor, TaWRKY42-B, facilitates initiation of leaf senescence by promoting jasmonic acid biosynthesis." BMC Plant Biology 20, no. 1 (2020): 1-22.

Author Response

Flag leaf senescence is an important determinant of wheat yield, as leaf senescence occurs in a coordinated manner during grain filling. We modified the text as suggested. Some studies investigated the leaf senescence under drought or different nitrogen fertilization rates or by wheat cDNA-array. Our study using the high-throughput sequencing technology to characterize possible senescence related genes at different development stages of flag leaves of wheat.

Round 2

Reviewer 1 Report

Authors have endorsed most of the comments and the quality of the manuscript is better in the new version. However, there are minor typo error and minor comments that need to be addressed. 

L60. It is the first time that authors mention Ca. Perhaps a different linker would be more appropriate. 

L114. To sum up, leaf senescence…

L116. reproductive organs. Although…

L116-118. Does this sentence make allusion to the references included in lines 119-125?

L147-149. Did crop receive 795 kg/ha of N as urea and diammonium phosphate? Are those rates common in that area?

L180. transcript. Prior

L187. Method (benjamini and Hochberg, 1995)

L188-189. Replace is by was. P-value

L227-228. The qRT-PCR was completed in the same three biological replications where the RNA-seq analysis was performed with three technical repeats.

L441. Traes_7DS_F5A240B02, Traes_2AL_8A23618BA,

L445.  DAA. The expression

L447. showed

L447. What do you mean by aeq-RNA?

L448. indicated… data (Fig S5).

L459. provide

L483. Do you mean that anabolism was stronger than catabolism in those leaves? Or that anabolism is always stronger than catabolism?

L494. Replace are by were

L523-524. Zhou et al., (2018) also reported that Rubisco might play a critical role in N deposition

L529. Replace is by was

L586-590. Gene names go in italic.

L624 DAA, the differential

L632 sensecence

Author Response

Dear Editors and Reviewers:

Thanks for your and for the reviewers’ comments concerning our manuscript entitled “Transcriptome Analysis of Early Senescence in the Post-Anthesis Flag Leaf of Wheat (Triticum aestivum L.)” (ID: 1851706). Those comments are all valuable and very helpful for revising and improving our paper, as well as the important guiding significance to our research. We have studied the comments carefully and have made corrections which we hope meet with approval. Revised portions were with the track changes. The main corrections in the paper and the responses to the reviewer’s comments are as flowing:

Responds to the reviewer’s comments:

Reviewer 1

  1. 1. It is the first time that authors mention Ca. Perhaps a different linker would be more appropriate.

Response: We made changes. Page 2 , L50-82.

  1. L114. To sum up, leaf senescence…

Response: Modified. Page 2 , L79.

  1. 3. L116. reproductive organs. Although…

Response: Modified. Page 3 , L134.

  1. 4. L116-118. Does this sentence make allusion to the references included in lines 119-125?

Response: No. L116-118 (Although there are many studies on senescence in wheat, arabidopsis, rice, maize, etc., most of them are based on the regulation of a single gene.). We want to summarize the research contents of the references included in lines L96-L124. Such as ZmNAC126, TaNAC29TaSNAC11-4B etc.

  1. L147-149. Did crop receive 795 kg/ha of N as urea and diammonium phosphate? Are those rates common in that area?

Response: The amount of fertilizer applied before sowing was 120 kg/ha urea and 300 kg/ha diammonium phosphate. Irrigation was carried out at the jointing stage of wheat, and 225 kg/ha of urea was applied. Wheat receives 645 kg/ha of N, P as urea and diammonium phosphate. Since this experiment does not involve the variable of fertilizer, conventional agricultural management was adopted. According to the local soil type and planting habits, the amount of fertilizer applied by local farmers throughout the growth period of wheat is about 500-600 kg/ha. Page 4, L177-180.

  1. L180. transcript. Prior

Response: Modified. Page 5, L212.

  1. L187. Method (benjamini and Hochberg, 1995)

Response: Modified. Page 5, L219.

  1. L188-189. Replace is by was. P-value

Response: Modified. Page 5, L220-222.

  1. L227-228. The qRT-PCR was completed in the same three biological replications where the RNA-seq analysis was performed with three technical repeats.

Response: Modified. Page 6, L259-260.

  1. L441. Traes_7DS_F5A240B02, Traes_2AL_8A23618BA,

Response: Modified. Page 13, L475.

  1. L445.  DAA. The expression

Response: Modified. Page 13, L479.

  1. L447. showed

Response: Modified. Page 13, L481.

  1. L447. What do you mean by aeq-RNA?

Response: RNA-seq. Page 13, L481.

  1. L448. indicated… data (Fig S5).

Response: Modified. Page 13, L481.

  1. L459. provide 

Response: Modified. Page 14 , L493.

16.L483. Do you mean that anabolism was stronger than catabolism in those leaves? Or that anabolism is always stronger than catabolism?

Response: Compared with 0–15 DAA, more metabolic pathways were enriched at 15–25 DAA and 25–30 DAA. Among them, more genes involved in the metabolism of fat, amino acids and carbohydrates are more up-genes in 15-25 DAA, then it is significantly down regulated in 25-30 DAA. After all, the wheat is in the filling stage when 15-25DAA, and photosynthesis synthesizes more assimilates rather than decomposes. Therefore, we believed that the anabolism of fat, amino acid and carbohydrate is stronger at 15-25 DAA (Fig.1). Anabolism was stronger than catabolism that it only represents 15-25 DAA, not always.

  1. L494. Replace are by were

Response: Modified. Page 15, L527.

  1. L523-524. Zhou et al., (2018) also reported that Rubisco might play a critical role in N deposition

Response: Modified. Page 15, L556.

  1. L529. Replace is by was

Response: Modified. Page 16, L563-580.

  1. L586-590. Gene names go in italic.

Response: Modified. Page 17, L619-623.

  1. L624 DAA, the differential

Response: Modified. Page 17, L658

  1. L632 sensecence

Response: Modified. Page 18, L666

Reviewer 3 Report

The authors have improved the MS structure, particularly the discussion section. However, the introduction is lengthy as it contains many unnecessary explanations. As the background literature suggests, plenty of research has been done to understand the leaf senescence process; the introduction should primarily focus on what is innovative in this study or how this study adds to the existing knowledge.     The L 129-132: “in this study, we used high-throughput sequencing technology to characterize possible senescence……” Authors should highlight what is new in this technique, how this analysis will likely add new information to the existing knowledge

Leaf senescence is a highly complex trait controlled by genetics and environment, while this research is conducted using a single wheat genotype and a one-year field study. Authors should justify how these data are reproducible; the senescence process may significantly differ if the growing environment changes.

The provided information about the growing environment is insufficient; it is difficult to explain the leaf-level responses based on the mean temperature alone. Leaf senescence responds to the maximum temperature during reproductive and grain-filling phases. Daily temperature and relative humidity data should be added – the results should be explained based on the specific environment and genotype used in this study.

Author Response

Dear Editors and Reviewers:

Thanks for your and for the reviewers’ comments concerning our manuscript entitled “Transcriptome Analysis of Early Senescence in the Post-Anthesis Flag Leaf of Wheat (Triticum aestivum L.)” (ID: 1851706). Those comments are all valuable and very helpful for revising and improving our paper, as well as the important guiding significance to our research. We have studied the comments carefully and have made corrections which we hope meet with approval. Revised portions were with the track changes. The main corrections in the paper and the responses to the reviewer’s comments are as flowing:

Reviewer 2

  1. 1. The authors have improved the MS structure, particularly the discussion section. However, the introduction is lengthy as it contains many unnecessary explanations. As the background literature suggests, plenty of research has been done to understand the leaf senescence process; the introduction should primarily focus on what is innovative in this study or how this study adds to the existing knowledge. The L 129-132: “in this study, we used high-throughput sequencing technology to characterize possible senescence……” Authors should highlight what is new in this technique, how this analysis will likely add new information to the existing knowledge.

Response: We tried to shorten the length of the introduction by combining the same statements and unnecessary explanations and made a lot of modifications to the introduction. The characteristics of high-throughput sequencing technology are added to L146-L157 to clarify the characteristics of sequencing depth and accuracy in this paper. With the progress of high-throughput sequencing equipment and technology, and the further improvement of the wheat genome. It can identify genes that have not been characterized or reported before, which is a supplement to the senescence database. Page 1-4, L42-158.

  1. 2. Leaf senescence is a highly complex trait controlled by genetics and environment, while this research is conducted using a single wheat genotype and a one-year field study. Authors should justify how these data are reproducible; the senescence process may significantly differ if the growing environment changes.

Response: Indeed, leaf senescence is a highly complex trait controlled by genetics and the environment. Some other researchers conducted experiments on environmental control in greenhouses. While we think it is important to conduct the experiment in the field, under the natural growing environment. One of the difficulties in field experiments is the control of the environment. We checked the weather conditions in the five years before and after the experimental year, and found no significant climate change. The change of environment may affect the change of aging degree and aging time, which further affects the change or advance of the expression of aging genes, and may rarely change the type of genes. Considering the cost of high-throughput sequencing, we did this experiment for one year. In order to save costs in high-throughput sequencing, there are also many articles that have only been sequenced for one year. The single genotype is used because the genome of ‘Chinese spring’ wheat has been sequenced and has a good annotation, which can provide more accurate experimental data.

  1. 3.The provided information about the growing environment is insufficient; it is difficult to explain the leaf-level responses based on the mean temperature alone. Leaf senescence responds to the maximum temperature during reproductive and grain-filling phases. Daily temperature and relative humidity data should be added – the results should be explained based on the specific environment and genotype used in this study.

Response: Temperature is one of the main external factors affecting senescence, so this is a very good suggestion, which is helpful for readers to better understand or apply the experimental data. Therefore, we obtained the meteorological data of the year of planting at Xianyang Meteorological Station in Shaanxi Province, and made a broken line chart of the temperature and relative humidity of wheat during the flowering-maturity period (Figure S1), and briefly described in the planting of materials. Page 4 , L173-182. Figure S1. Please see the attachment. If needed, we can attach all the weather data for every day.

Round 3

Reviewer 3 Report

The authors have substantially improved the MS quality, I have no more comments and the MS may be accepted for publication.